# Novel Assessment of Collagen and Its Crosslink Content in the Humerus from Primiparous Dairy Cows with Spontaneous Humeral Fractures Due to Osteoporosis from New Zealand

**DOI:** 10.3390/biology11101387

**Published:** 2022-09-23

**Authors:** Alvaro Wehrle-Martinez, Rafea Naffa, Penny Back, Chris W. Rogers, Kevin Lawrence, Trevor Loo, Andrew Sutherland-Smith, Keren Dittmer

**Affiliations:** 1School of Veterinary Sciences, Massey University, Private Bag 11-222, Palmerston North 4442, New Zealand; 2Fonterra Research and Development Centre, Palmerston North 4472, New Zealand; 3School of Agriculture and Environment, Massey University, Private Bag 11-222, Palmerston North 4442, New Zealand; 4School of Natural Sciences, Massey University, Private Bag 11-222, Palmerston North 4442, New Zealand

**Keywords:** undernutrition, collagen, collagen crosslink, fracture, dairy cows, copper deficiency

## Abstract

**Simple Summary:**

Collagen and collagen crosslinking are important determinants of bone strength. For the first time, collagen and its crosslink content were measured in the humerus from primiparous dairy cows with spontaneous humeral fracture due to osteoporosis from New Zealand. The results were compared with the humerus from unfractured dairy cows. Cows with a humeral fracture have a lower total collagen content but higher total collagen crosslink content compared with unfractured cows. The results suggest that protein/calorie undernutrition might be an important factor associated with the incidence of spontaneous humeral fractures in dairy cows from New Zealand.

**Abstract:**

Numerous cases of spontaneous humeral fracture in primiparous dairy cows from New Zealand have prompted the study of the condition to establish probable causes or risk factors associated with the condition. Previous studies identified inadequate protein-calorie malnutrition as an important contributory factor. Earlier case studies also reported that ~50% of cows have low liver and/or serum copper concentration at the time of humeral fracture. Because copper is so closely associated with the formation of collagen cross-links, the aim of this study was to compare collagen and collagen crosslink content in the humerus from primiparous cows with and without humeral fractures and to determine the role of copper in the occurrence of these fractures. Humeri were collected from cows with and without humeral fractures, ground, and the collagen and collagen cross-link content measured using high-performance liquid chromatography. Collagen content was significantly higher in the humeri of cows without humeral fractures, while total collagen crosslink content was significantly higher in the humerus of cows with humeral fractures. These results indicate other factor/s (e.g., protein-calorie undernutrition) might be more important than the copper status in the occurrence of humeral fractures in dairy cows in New Zealand.

## 1. Introduction

Since the first reported case of spontaneous humeral fracture in primiparous dairy cows in New Zealand, several risk factors have been hypothesized as associated with an increased incidence of fractures [1,2,3]. The largest histological study on humeral fractures determined that cows have osteoporosis due to inadequate bone deposition during the first months of the cow’s life and increased bone resorption in the post calving period [4]. Another study, using peripheral quantitative computed tomography, has shown a consistent trend for reduced bone length and reduced cortical bone mineral density in primiparous cows with humeral fractures compared with age-matched controls [3]. The findings from both these studies support the hypothesis that periods of inadequate nutrition (most likely protein-calorie undernutrition) act as a contributory factor in the occurrence of humeral fractures in cows and furthermore suggest these changes are recent [3,4].

Another relatively common finding from previous studies is that many cows affected with humeral fractures have either low liver and/or serum copper (Cu) concentrations [2,3,5]. For example, in the very first reported outbreak of humeral fractures in dairy cows, Cu deficiency was diagnosed and associated with the outbreak [5]. A follow-up larger study also reported low serum and/or liver Cu concentration in some of the affected animals and proposed Cu deficiency as a contributing factor, which weakened bone, and contributed to osteoporosis and subsequent humeral fracture [2].

Copper deficiency is one of the most common trace mineral deficiencies in cattle in New Zealand [6]. Secondary Cu deficiency is more common than primary deficiency and is associated with excessive concentrations of Cu antagonists (including molybdenum (Mo), sulphur, iron (Fe), zinc (Zn), and manganese) in the diet and/or soil that can reduce Cu bioavailability, absorption, and cellular use [6,7,8].

The relationship between Cu and bone strength is related to the activity of lysyl oxidase (LOX), a pivotal copper-dependent enzyme in the formation of collagen crosslinks [9,10]. Collagen crosslinking contributes to the mechanical properties of bone and LOX oversees the most important step in the formation of collagen crosslinks, which is the conversion of specific lysine ε-amino groups and the stabilization of the triple helical collagen molecule [11,12]. Collagen crosslinks can be classified as either immature or mature crosslinks (among other types) [13]. Immature collagen crosslinks convert spontaneously to mature collagen crosslinks, and normal human healthy bone is characterized as having 2–4 times the content of immature compared to mature collagen crosslinks [13]. As a result of Cu deficiency (primary or secondary), changes in the amount and type of collagen crosslinking can occur [13,14,15]. This can impact bone mechanical properties and strength, and as a result, Cu deficiency has been associated with increased bone fragility and an increased incidence of spontaneous bone fracture, among other clinical signs [9,16,17].

The objectives of this study were (1) to quantify and describe any significant alterations in collagen and collagen crosslink content in humeri from cows with humeral fractures due to osteoporosis, compared with age-matched post calving cows without humeral fracture; (2) to assess the relationship between liver and bone Cu concentration and collagen and collagen crosslinking in bone; and (3) to compare the liver concentration of cadmium (Cd), Fe, Mo, and Zn (Cu antagonists) between cows with and without humeral fracture, to determine if Cu antagonists are causing secondary Cu deficiency.

## 2. Materials and Methods

### 2.1. Study Design and Sample Collection

This was a case–control study using a convenience sample of humeral cortical bone from primiparous cows with osteoporosis that fractured one or both humeri (affected group) and non-fractured animals (control group). The case definition for enrolling an animal in the affected group was a dairy cow of any breed, at least 2 years old, which had suffered a spontaneous fracture of the humerus, without any history of trauma, within 6 months of calving. The case samples were provided by farmers and veterinarians around New Zealand who after reporting a case of spontaneous fracture of the humerus were asked to collect a sample of the humerus and a piece of liver postmortem. In this group, blood samples were also collected by the referring veterinarian as part of a diagnostic investigation into the cause of the fracture. The collection of samples occurred between July and December 2019.

Control samples were obtained from a cow rendering plant (Wallace Corporation, Feilding, New Zealand) and Massey University School of Veterinary Science postmortem service. Samples for control cases were taken from dairy cows of any breed, with an ear tag indicating they were at least 2 years old, who had calved recently (udder consistent with lactating) and had been culled for reasons unrelated to bone fracture of the humerus or any other bone. From each control animal, a sample of the humerus and a piece of liver was collected postmortem. Antemortem blood samples were, however, not available owing to the method of sampling.

For both groups (affected and control), no information regarding sample handling and/or time between collection and reception was reported and/or recorded in this study. Most samples from affected cows were sent overnight by courier.

### 2.2. Preparation of Bone Samples

A total of 26 humeri from affected cows and 14 control humeri were collected. From each, bone slabs (from the proximal humeral epiphysis and metaphysis) were obtained using an industrial-grade band saw. One bone slab was cleaned with high-pressure cold water to remove the bone marrow. From each slab, cortical bone was dissected out and ground using a cryogenic grinder filled with liquid nitrogen (6875 Freezer/Mill^®^, SPEX^®^ SamplePrep, Metuchen, NJ, USA). The grind protocol included two cycles each with a pre-cool step, a run step of 2 min, a cool step, and a second run time of 2 min. The impactor rate was set at 5 cycles per second. Once the cycle was finished, powdered material was stored in Eppendorf tubes covered with aluminum foil at −80 °C until further processing.

### 2.3. Collagen and Collagen Crosslink Analysis of Cow Bone

#### 2.3.1. Materials

Sodium hydroxide, sodium borohydride 98%, chloramine T trihydrate, and 4-(dimethylamino) benzaldehyde were purchased from Sigma-Aldrich (St. Louis, MO, USA). Mass spectrometry grade water, acetonitrile (MeCN), formic acid, methanol, acetic acid glacial, hydrochloric acid S.G 1.18 (~37%), and sodium hydroxide were sourced from Fisher Scientific (Fair Lawn, NJ, USA). Perchloric acid 70% and propan-1-ol were purchased from BDH Chemicals Limited (Poole, England). Deionized water was obtained from a Milli-Q Ultra-pure water system (Dubuque, IA, USA). Dihydroxylysinonorleucine standard (DHLNL) was purchased from Santa Cruz Biotechnology (Santa Cruz, CA, USA). Pyridinoline (PYD) (95%) and deoxypyridinoline (DPD) (98%) standards were purchased from BOC Science (New York, NY, USA). Hydroxylysinonorleucine (HLNL), histidinohydroxylysinonorleucine (HHL), histidinohydroxymerodesmosine (HHMD) standards, were isolated and purified in the Protein Structure and Function Laboratory, Massey University’s School of Natural Sciences [16].

#### 2.3.2. Bone Powder Preparation

Around 100–150 mg of powdered cortical bone sample from each case (26 affected cows and 14 control cows) was freeze-dried for 24 h. For this, Eppendorf tubes with powdered material were placed in a stainless-steel stockpot, with their lids open but covered with parafilm^®^ sealing film with multiple holes in it. The stockpot was closed-sealed and connected to the freeze dryer (SP VirTis Freezemobile 35EL Freeze Dryer, SP equipment, Warminster, PA, USA). Bone samples were then reduced using sodium borohydride (NaBH_4_) as previously described [17]. Briefly, 10 mg of NaBH_4_ was dissolved in 1 mL of 1 mM cold sodium hydroxide to give a 1:10 ratio of sodium borohydride to bone powder, which was subsequently added to the bone samples. The mixture was incubated for 24 h at 37 °C. The reduction was stopped using glacial acetic acid, added until the pH dropped to 3.0. Samples were then centrifuged at 5000× *g* for 5 min at 19 °C and the supernatant was discarded. The pellets were then washed with 5 mL of distilled water, to remove excess acetic acid and salts, centrifuged again (5000× *g* for 5 min at 19 °C), supernatant discarded, and freeze-dried overnight.

The following day samples were first weighed and then acid hydrolyzed using 1.5 mL of 6 M HCl at 105 °C for 24 h. After that, to neutralize the HCl in the resulting hydrolysate, 2.5 mL of 6 M NaOH was added to the sample, mixed, and filtered through glass-wool plugged plastic syringes. Two filtrations were carried out using 2.5 mL of distilled water, samples were then frozen in liquid nitrogen, connected to the freeze dryer, and freeze-dried for 24–48 h. Samples were then stored at −80 °C until further testing.

#### 2.3.3. Crosslink Separation and Quantification

To quantify the different collagen crosslinks in bone powder samples from affected and control cows, three biological replicates were used, with each biological replicate being tested for extraction efficiency using three technical replicates. Before crosslink separation and quantification, samples of the prepared bone powder were rehydrated in 1000 µL of water and 200 µL of this was inserted into a 1.5 mL short thread vial (Thermo Scientific, Waltham, MA, USA). Separation of collagen crosslinks was performed, as previously reported, by liquid chromatography (LC) using a Dionex UltiMate^TM^ 3000 system with an autosampler and a LPG-3400RS Rapid Separation Quaternary Pump, Thermo Fisher Scientific, USA) with a Cogent Diamond Hydride™ HPLC column (2.2 µm, 100Å, 150 mm × 2.1 mm ID, PM Separations, Capalaba, Queensland, Australia) [17]. The LC system was coupled to a Q Exactive^TM^ Focus mass spectrometer equipped with a high-energy collision-induced dissociation collision cell, an Orbitrap mass analyzer, and a HESI-II ion source (Thermo Fisher Scientific, USA) for crosslink quantification. Parallel reaction monitoring using tandem mass spectrometry acquisition with an inclusion list of ions was used to detect and quantitate the relevant ions for each crosslink. Details for the chromatographic and mass spectrometry settings are listed in Appendix A.

#### 2.3.4. Determination of Total Collagen Content in Cortical Bone

The amount of collagen in the prepared bone powder was determined using a hydroxyproline (HYP) assay with an aliquot of the acid hydrolysate as described by Reddy and Enwemeka [18]. In summary, aliquots of the standard HYP (2–20 µL) prepared from stock solution (1 mg/mL of HYP in water) and the bone powder hydrolysate (at different volumes; 2.5 to 10 µL) were mixed with water in a total volume of 50 µL. Next, 450 µL of chloramine-T was added, mixed gently, and incubated for 25 min at room temperature. Then, 500 µL of Ehrlich’s aldehyde reagent was added, and the sample mixed and incubated at 65 °C for 20 min for the chromophore reaction to develop. Finally, absorbance was read at 550 nm using a microplate spectrophotometer (BioTek PowerWave XS, Winooski, VT, USA). Collagen content was calculated on the dry weight of the bone assuming 14% HYP in type I collagen [19].

#### 2.3.5. Determination of the Copper Concentration (in Bone, the Liver and Serum) and the Liver Concentration of Copper and Copper Antagonists (Cadmium, Iron, Molybdenum, and Zinc)

The same samples from affected (*n* = 26) and control (*n* = 14) cows that were used for the determination of collagen and collagen crosslink concentration were also used for determination of Cu (in the bone, the liver and serum) and determination of the liver concentration of Cu antagonists. Around 300 mg of bone power from each case (*n* = 40, cortical and trabecular bone) was prepared and analyzed for bone Cu concentration at Gribbles Scientific (Mosgiel, New Zealand). Liver samples (*n* = 40) were collected and analyzed for Cu concentration at IDEXX Laboratories Pty. Ltd. (Palmerston North, New Zealand) and analyzed for Cd, Fe, Mo, and Zn at Gribbles Scientific (Mosgiel, New Zealand). The liver Cu concentration was categorized as low (0–94 µmol/kg) or adequate (>94 µmol/kg) [20].

Blood samples were only available for cows in the affected group (*n* = 26). Upon reception, samples were centrifugated at 1008× *g* for 15 min and analyzed for serum Cu concentration at IDEXX Laboratories Pty. Ltd. (Palmerston North, New Zealand). Serum Cu concentration was categorized as low (0–7.9 µmol/L) or adequate (>7.9 µmol/L) [20].

#### 2.3.6. Data Analysis

Mass spectrometry data were processed using Xcalibur^TM^ Software v.4.1.31.9 (Thermo Scientific^TM^, Waltham, MA, USA). The FreeStyle^TM^ Software v.1.3.115.19 (Thermo Scientific^TM^, Waltham, MA, USA) was used for data visualization. TraceFinder™ 4.1 version 4.1.265.0 software was used to extract and quantify the ion peaks, and the results were exported into Excel for quantification.

Statistical analysis was done on the total collagen content and, DHNLN, PYD, and DPD crosslinks. Before statistical analysis, all values were normalized to the HYP content in bone (total crosslink/total HYP in sample) [17]. Each bone crosslink result, for fractured and control bones, was the mean of three biological replicates, each the mean of their respective three extraction technical replicates.

An independent-samples t-test was used to determine if there were any significant differences between Cu concentration (in the liver, serum, and bone), the liver Cd, Fe, Mo, and Zn concentration, the total crosslink content (DHLNL + PYD + DPD), total collagen content, DHLNL, PYD, and DPD content and DHLNL/(PYD + DPD) ratio in bones from affected and control cows. Values were first natural log transformed to achieve a normal distribution. Results are back transformed and presented as mean ± standard deviation unless otherwise stated. A *p* value of <0.05 was considered significant. If the assumption of homogeneity of variances was violated a Welch t-test was run to determine differences between groups.

A Spearman’s rank-order correlation test (results displayed as *r_s_*) was used to determine the strength and direction of the relationship between liver and bone Cu concentration, total collagen, DHLNL, PYD, and DPD content in bone. A *p* value of <0.05 was considered significant. A point-biserial correlation test (results displayed as *r_pb_*) was used to determine the strength and direction of the relationship between liver and bone Cu concentration (as the continuous variable) and fracture status (affected and control) as a dichotomous variable.

Next, a simple linear regression test was built to determine the relationship between liver or bone Cu concentration (as the predictor variable) based on the value of DHLNL, PYD, and DPD collagen crosslinks and total collagen content in bone (outcome variables).

Finally, two binomial logistic regression models were built. The first, to determine the effect of liver and bone Cu concentration on the odds of a cow having a humeral fracture and the second, to determine the effect of the quantity of collagen crosslinks on the odds of a cow having a humeral fracture. For a variable to be included as a parameter in the logistic model it had to have an observed count (distribution) of more than 1, and the *p* value obtained from the t-test a *p* value < 0.25 [21]. The model was constructed using forward selection, with variables retained at *p* < 0.05, and the model fit was assessed using the Hosmer-Lemeshow (HL) test and the Nagelkerke R2. All statistical analyses were performed using SPSS statistics (IBM^®^ SPSS^®^ Statistics version 27).

## 3. Results

A total of 40 bone cortices and liver samples from 26 affected cows and 14 control cows were used in this study.

### 3.1. Copper Concentration

The mean values of the liver and bone Cu concentration are shown in Table 1. The mean liver Cu concentration was adequate in affected and control cows, although significantly higher in control cows (*p* < 0.001). The mean serum Cu concentration in affected cows was 15.8 ± 1.42 µmol/L. Bone Cu concentration was significantly higher in affected cows compared with control cows (*p* < 0.001). The correlation between liver (*r_pb_*(38) = 0.504, *p* < 0.001) and bone Cu (*r_pb_*(38) = −0.582, *p* < 0.001) concentration and fracture status was found to be statistically significant, with affected cows having the lowest liver Cu concentration and the highest bone Cu concentration. Similarly, the correlation between serum and liver Cu concentration in affected cows was found to be statistically significant and positive (*r_s_*(38) = 0.39, *p* = 0.05). Other correlations did not provide significant results.

The following two parameters were tested in the logistic regression model: liver and bone Cu concentration. Both were retained in the final model. The logistic regression model was statistically significant, χ^2^(4) = 21.56, *p* < 0.001. The Hosmer and Lemeshow test was not statistically significant (*p* = 0.30) indicating the model was a good fit. The model explained 58.1% (Nagelkerke R2) of the variance in humeral fracture and correctly classified 85% of cases. Sensitivity was 69.2% and specificity was 92.6%. Increased bone Cu concentration was associated with increased odds (56.1 times) of having a humeral fracture while for decreased liver Cu concentration, the odds of having a humeral fracture increased by a factor of 3.2 (Table 2).

### 3.2. Collagen Content

As shown in Table 3, the total collagen content was significantly higher in control cows compared with affected cows (*p* = 0.004). A significant correlation was found between bone Cu concentration and total collagen content in bone (*r_s_*(38) = −0.45, *p* = 0.004.). Other correlations did not provide significant results.

### 3.3. Collagen Crosslinks Concentration

Bone samples did not contain HHMD crosslinks and the quantity of HLNL was either not found or extremely low. All bone samples analyzed contained DHLNL, DPD, and PYD collagen crosslinks.

Table 4 provides the summary statistics for collagen crosslinks. The total crosslink content (DHLNL + PYD + DPD) was significantly higher in affected cows and cows with low liver Cu concentration compared with control cows (*p* < 0.001) and cows with adequate liver Cu concentration (*p* = 0.007). Bone samples from affected cows contained significantly more DHLNL and PYD crosslinks compared with control cows (*p* = 0.009 and *p* = 0.002, respectively). Bone samples from cows with low liver Cu concentration had significantly higher DHLNL and PYD compared with cows that had adequate liver Cu concentration (*p* = 0.045 and *p* = 0.030, respectively).

A significant correlation was only found between bone Cu concentration and the quantity of DHLNL in bone (*r_s_*(38) = −0.38, *p* = 0.017). Other correlations did not provide significant results. Linear regression analysis showed liver Cu concentration explained very little of the variation in crosslink concentration. In contrast, linear regression showed bone Cu concentration could statistically predict DHLNL and HYP concentration (*F*(1, 38) = 4.21, *p* = 0.047 and *F*(1, 38) = 5.81, *p* = 0.021, respectively. Bone Cu concentration accounted for 10% of the variation in DHLNL concentration and, 13.3% of the variation of HYP concentration in bone.

The following three parameters were tested in the logistic regression model: HYP, DHLNL, and PYD. DHLNL and PYD were retained in the final model. The logistic regression model was statistically significant, χ^2^(4) = 16.81, *p* < 0.001. The Hosmer and Lemeshow test was not statistically significant (*p* = 0.63) indicating the model is a good fit. The model explained 47.9% (Nagelkerke R2) of the variance in humeral fracture and correctly classified 77.5% of cases. Sensitivity was 85.2% and specificity was 61.4% (Table 5).

### 3.4. Copper Antagonists

There were no significant differences in the mean liver concentration of Mo, Zn, Fe, and Cd between affected and control cows, and between cows with low or adequate liver Cu concentration.

## 4. Discussion

The results of this study indicate that in cows with osteoporosis and humeral fractures low liver Cu concentration is correlated with the presence of a humeral fracture, indicating there is significant mobilization of liver Cu. This depletion has seemingly resulted from Cu being transported to extrahepatic tissues including bone, with bone Cu concentration being significantly higher in affected cows than in control cows and correlated with the presence of humeral fracture. To the best of our knowledge, this is the first time such a finding is described. Furthermore, these findings indicate that bone Cu concentrations explain a greater amount of the variability in the presence of fracture in cows compared with liver Cu concentration (33.9% vs. 28.2%) and increased bone Cu concentration is associated with a far greater risk of having an osteoporotic humeral fracture (56.1 times more) compared to low liver Cu concentration (only 3.2 times more).

For the first time, we were able to measure the total collagen content and collagen crosslink content in bone from dairy cows in New Zealand which allowed us to establish important conclusions regarding the likely pathogenesis of spontaneous humeral fracture. In the case of collagen and its crosslinks, the total collagen content was significantly higher in bone from control cows and cows with adequate liver Cu concentration consistent with the histologic findings of osteoporosis reported [4]. Cows with humeral fracture and low liver Cu concentration have an increased concentration of total collagen crosslinks, DHLNL, and HYP in bone. Although only bone Cu concentration was positively correlated with DHLNL. The concentration of Cu in the liver does not correlate with any collagen crosslinks indicating no correlation between these two components.

Since the first reported case of humeral fracture, low liver Cu or serum Cu concentration has been a common finding in cows and/or herds with spontaneous humeral fractures [1,2,5]. After intestinal absorption, Cu is stored in the liver and liver Cu concentration is measured to evaluate the Cu status in cows. Finding liver and/or serum Cu concentration below the reference range indicates deficient Cu consumption, absorption, loss of Cu storage pool and Cu deficiency [8]. The reported low liver or serum Cu concentration in cases of humeral fractures has been suggested to be a potential cause of bone weakness leading to fracture This association comes from the fact that Cu deficiency, either alone or with other nutritional deficiencies, can result in musculoskeletal disorders (including lameness, enlargement of joints, radiographical loss of cortical bone index) and bone fractures in both animals and humans [6,22,23,24,25]. The relationship between Cu and bone quality/strength is linked to the activity of the Cu-dependent lysyl oxidase (LOX) enzyme [13,26]. This enzyme has a central role in the control of the total amount of collagen crosslinking, an important determinant of bone quality [13].

Like many previous reported case studies, most affected cows in this study (16/26, 62%) had low liver Cu concentration (<94 µmol/kg) at the time of fracture [1,2,5,27,28,29,30]. The concentration of Cu in the liver is defined as the Cu “storage pool” and indicates that in affected cows there was inadequate Cu consumption for at least 30 days before testing, with subsequent mobilization of stored Cu from the liver [8].

What previous studies have not been able to determine, is whether the liver Cu deficiency equates to a whole-body Cu deficiency and the destination of the Cu that was stored in the liver. Serum Cu concentration, only measured in affected cows, revealed that most animals (22/26, 85%) had adequate serum Cu concentration (>7.9 µmol/L) and furthermore, liver Cu and serum Cu concentration were significantly correlated within the affected group. This finding indicates that in affected cows, although Cu storage is being depleted, there is still enough Cu to maintain adequate serum Cu concentrations (known as the Cu transportation pool).

The next step is to evaluate Cu concentration in the tissues where it is needed; this is regarded as the functional or tissue pool [8]. Experimental Cu deficiency in calves did not reduce bone Cu concentrations when compared with Cu-supplemented calves (measured in bone ash using atomic absorption spectrophotometry) [25]. In contrast, chicks on a Cu deficient diet had lower bone Cu content compared with chicks on a Cu-supplemented diet [15]. However, cows with humeral fractures and cows with low liver Cu concentration had significantly higher bone Cu concentration compared with control cows and cows with adequate liver Cu concentration. Although bone and liver Cu concentration are not significantly correlated in this study, the liver Cu depletion appears to be linked to an increase in the bone Cu concentration (and possibly other tissues), probably in response to a stimulus from the bone. Additionally, increased bone Cu concentration (OR 56.1) and not decreased liver Cu concentration (OR 3.2) has a greater association with humeral fracture. Therefore, it seems likely that the low liver Cu concentration, so frequently described in cows with humeral fractures is, at least in some part, due to mobilization of Cu storage to bone and not indicative of a clinical Cu deficiency in these cows. The increase in bone Cu concentration and not the decrease in liver Cu concentration is more closely associated with humeral fracture in cows in New Zealand.

Collagen stabilization in the extracellular matrix is dependent on the formation of inter- and intramolecular crosslinks between collagen fibers [12]. This is known as the enzymatic pathway of collagen crosslinking and results in the formation of immature crosslinks (such as DHLNL, measured in this study) [26]. The next step is the conversion of immature crosslinks to mature crosslinks, such as PYD and DPD (both measured in this study) [26].

In bone, the conversion of immature to mature crosslinks is described as a continuous, and independent process [12,13]. In normal bone, immature crosslink content is higher than mature crosslinks and bone is considered to be one of the only tissues to contain a significant pool of immature crosslinks (2–4 times the content of mature crosslinks) [12,13]. When comparing the content of immature (DHLNL) and mature (PYD) collagen crosslinks in bones from control cows and cows with adequate liver Cu concentration, DHLNL was 3.1 and 2.9 times more than PYD, consistent with the ratio of immature to mature crosslinks reported in the literature for healthy bone [12,13]. Similar ratios are observed comparing DHLNL and PYD in bones from affected cows and cows with low liver Cu concentration.

Although the conversion (immature to mature crosslinks) is continuous, bone growth and the bone turnover rate can affect the relative amounts of immature and mature crosslinks [13]. In Cu deficiency there is a reduction in the activity of LOX which impairs collagen crosslinking [13]. For example, in Cu-deficient chicks (due to low dietary Cu) low LOX activity was related to decreased immature crosslink concentration (mainly DHLNL) and a reduction in bone torsional strength [14]. In a study of osteoporotic avian bone, while there were no significant differences in the content of DHLNL compared with normal bone, decreased DHLNL was correlated with a decrease in breaking strength [31]. Finally, in postmenopausal osteoporosis in women there is a significant reduction in immature crosslink concentration [13].

Contrary to these previously mentioned studies, the current study found significantly higher content of immature (DHLNL) collagen crosslink in bones from affected cows (1.23 times more) and cows with low liver Cu concentration (1.20 times more) and additionally, bone Cu concentration was correlated with DHLNL amount. Notably, mechanical stress can accelerate the conversion of immature to mature crosslinks as an attempt to stabilize the collagen molecule and this is controlled by LOX [13]. This is likely to be occurring in the bones of cows with humeral fractures considering the bone in these cows is characterized by a reduction in trabecular density, abnormal trabecular architecture, and thinner cortex which can significantly compromise bone biomechanical strength [32,33]. Furthermore, high DHLNL concentration in bone is indicative of a higher collagen turnover rate and could lead to thinner newly synthesized collagen molecules that alter the properties of bone [31].

Similarly, PYD (mature crosslink) was significantly higher in affected cows (1.5 times more) and cows with low liver Cu (1.23 times more). As with DHLNL, reduced LOX activity inhibits the formation of PYD crosslinks which reduces bone strength [12,13]. The increased content of mature crosslinks (PYD and DPD) in bones from affected cows and cows with low liver Cu concentration is likely in response to the higher concentration of DHLNL previously described and further support the hypothesis of a higher conversion rate of crosslinks to allow extra stabilization of the collagen molecule in cows with humeral fractures.

The total collagen content was also measured in cows. In chicks on Cu-deficient diets, the content of soluble collagen in bone significantly increased compared with chicks on Cu-supplemented diets and this was thought to be related to decreased crosslinking (fewer crosslinks allowed greater ease of solubilization) and decreased mineralization (demineralization of bone before extraction solubilizes collagen hence Ca may protect from solubilization) [15]. Similarly, an increase in collagen synthesis and production was found in the aorta of pigs on a Cu-deficient diet [34]. Our findings suggest that another factor may have influenced collagen synthesis in affected cows causing a reduction in the production of collagen such as described here. In undernourished cattle, there is a significant reduction in serum albumin and creatinine concentration compared to adequately nourished animals [35]. We have previously shown that 69% of affected cows have low creatinine concentrations which was linked to low muscle mass and protein/calorie undernutrition [36]. Likely, undernutrition and/or low muscle mass and/or low body condition score may explain the lower total collagen content in bone from affected cows in this study.

Some limitations that should be acknowledged in this study include the number of control cases and the fact that all control cases had liver Cu concentration within the reference range which might not be indicative of the true Cu status for control cow population. Additionally, Cu concentration was measured at one point in time (the time of fracture) as such Cu status in the months prior to fracture is unknown. To the authors knowledge there is no information regarding changes in bone and/or liver Cu concentration during cow growth, gestation, and lactation.

Finally, we measured the concentration of known Cu antagonists, Cd, Fe, Mo, and Zn in the liver. As with Cu, the liver is the major organ of storage for these elements and reflects consumption levels [37]. Significant accumulation of Cd, Mo, Zn, and Fe were not found suggesting a minimal impact on low liver Cu concentrations in cows with humeral fracture.

## 5. Conclusions

This study was intended to determine the true role of collagen crosslinking and Cu as a determinant of bone strength in cows with a humeral fracture in New Zealand. The low liver Cu concentration found in most affected cows is likely due to the mobilization of Cu to other tissues (including bone). The high content of immature (correlated with bone Cu concentration) and mature collagen crosslinks is likely a response mechanism from a biomechanically stressed bone intended to improve the stabilization of the collagen molecule in cows with osteoporosis.

In conclusion, Cu deficiency is not a major factor in the pathogenesis of humeral fractures in dairy cows in New Zealand. The low total collagen content could indicate that inadequate feed quality/quantity is more important in the pathogenesis of this condition, thus increasing the incidence of spontaneous humeral fractures.

## Figures and Tables

**Table 1 biology-11-01387-t001:** Mean ± sd of liver and bone copper concentration in cows comparing humeral fracture status (affected vs. control) and mean ± sd bone copper concentration comparing low vs. adequate liver copper concentration; ** Welch *t*-test.

	Case	Mean ± sd	*p* Value
Liver copper (µmol/kg )	Affected (n = 26) Control (n = 14)	237.6 ± 385.3 468.0 ± 285.3	<0.001 **
Bone copper (mg/kg)	Affected (n = 26) Control (n = 14)	0.69 ± 0.27 0.41 ± 0.12	<0.001
	Low LivCu (n = 16) Adequate LivCu (n = 24)	0.68 ± 0.28 0.54 ± 0.24	0.07

**Table 2 biology-11-01387-t002:** Results of the logistic regression test for predicting humeral fracture in cows based on liver and bone Cu concentration; *B*: B coefficient for the constant; SE: standard error; Wald: Wald chi-square test; *df*: degrees of freedom; CI: confidence interval; Cu: copper; LivCu: liver copper concentration.

	*B*	SE	Wald	*df*	*p*	Odds Ratio	95% CI for Odds Ratio	
							Lower	Upper
Bone Cu	0.40	1.55	6.75	1	0.01	56.1	2.7	1168.9
LivCu	1.17	0.46	6.33	1	0.01	3.2	1.3	8.1
Constant	9.79	3.30	8.78	1	0.003			

**Table 3 biology-11-01387-t003:** Mean ± sd of total collagen content in cows comparing humeral fracture status (affected vs. control) and the liver copper concentration status (low vs. adequate); LivCu: liver copper concentration.

	Case	Mean ± Sd	*p* Value
Total collagen (mg)	Affected (n = 26) Control (n = 14)	1.71 ± 0.43 2.18 ± 0.46	0.004
	Low LivCu (n = 16) Adequate LivCu (n = 24)	1.79 ± 0.44 1.93 ± 0.53	0.463

**Table 4 biology-11-01387-t004:** Mean ± sd of total collagen crosslink content, DHLNL: dihydroxylysinonorleucine, PYD: pyridinoline, DPD: deoxypyridinoline, and DHLNL/(PYD + DPD) comparing humeral fracture status (affected vs. control) and liver Cu concentration status (low vs. adequate); LivCu: liver copper concentration. The crosslinks were normalised to the total HYP content (total crosslink in sample/total HYP in sample).

	Case	Mean ± sd	*p* Value
Total crosslink	Affected (n = 26) Control (n = 14)	3979.07 ± 797.72 3058.71 ± 548.70	<0.001
	Low LivCu (n = 16) Adequate LivCu (n = 24)	4082.32 ± 859.21 3373.36 ± 711.16	0.007
Immature crosslink			
DHLNL(mg/mol)	Affected (n = 26) Control (n = 14)	2847.23 ± 717.27 2257.38 ± 489.29	0.009
	Low LivCu (n = 16) Adequate LivCu (n = 24)	2943.09 ± 835.07 2439.25 ± 519.41	0.04
Mature crosslinks			
PYD(mg/mol)	Affected (n = 26) Control (n = 14)	1043.89 ± 403.18 724.98 ± 159.41	0.002
	Low LivCu (n = 16) Adequate LivCu (n = 24)	1049.00 ± 334.29 854.45 ± 377.64	0.03
DPD(mg/mol)	Affected (n = 26) Control (n = 14)	87.95 ± 25.75 76.34 ± 23.13	0.16
	Low LivCu (n = 16) Adequate LivCu (n = 24)	90.23 ± 26.07 79.65 ± 24.22	0.19
Immature/mature crosslinks			
DHLNL/(PYD + DPD) (mg/mol)	Affected (n = 26) Control (n = 14)	2.77 ± 1.02 2.92 ± 0.67	0.46
	Low LivCu (n = 16) Adequate LivCu (n = 24)	2.81 ± 1.12 2.83 ± 0.76	0.77

**Table 5 biology-11-01387-t005:** Results of the logistic regression test for predicting humeral fracture in cows based on PYD: pyridinoline and DHLNL: dihydroxylysinonorleucine content in bone; *B:* B coefficient for the constant; SE: standard error; Wald: Wald chi-square test; *df*: degrees of freedom; CI: confidence interval.

	*B*	SE	Wald	*df*	*p*	Odds Ratio	95% CI for Odds Ratio	
							Lower	Upper
PYD content	0.002	0.001	4.259	1	0.04	1.002	1.000	1.003
DHLNL content	0.006	0.003	4.385	1	0.04	1.006	1.000	1.012
Constant	−8.314	3.134	7.038	1	0.008			

## Data Availability

Data sharing not applicable.

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
