# Peer review of "Novel Assessment of Collagen and Its Crosslink Content in the Humerus from Primiparous Dairy Cows with Spontaneous Humeral Fractures Due to Osteoporosis from New Zealand"

_biology, 2022, doi:10.3390/biology11101387_

Round 1

Reviewer 1 Report

Minor or trivial comments mainly, the ms is clearly written.

line 21, trivial point but 'compared with' is more appropriate here than 'compared to', according to standard dictionaries

line 25 'increased incidence...' increased over what? suggest rephrase.

line 44 you don't need both 'putative' and 'hypothesised', one of them is redundant

line 55 commas either side of however. '.....,however,.... '

line 319 'affected fracture status', please clarify

line 335, comma before although: ...in bone, although....'

line 354 suggest replace 'describe' by 'determine'

General comments on discussion

1. a couple of sentences outlining normal copper metabolism in cattle would help the reader not conversant with this

2. Could the discussion be broadened a little by comparisons with collagen cross-linking in other osteoporotic situations (in human, for example)?

3. relating to the above point, what do the hormonal environments of pregnancy and lactation do to collagen and its cross-linking, is anything known on that in cattle? are there osteoporotic fracture models unrelated to pregnancy and lactation in cattle or other mammals that could through any light on this topic?

Reviewer 2 Report

I read Wehrle-Martinez et al article which title of " Novel assessment of collagen and its crosslink content in the humerus from primiparous dairy cows with spontaneous humeral fractures due to osteoporosis from New Zealand" carefully. The paper's excellent writing, well planned research, and addition of important scientific knowledge to the literature left me with a lasting impression.

The study's objectives, methodology, and conclusions are all clearly laid out in the abstract section, which also incorporates pertinent data.

The introduction is clear and concise, and the facts and methodology are both clearly described. The purpose is stated in the well-written in introduction section.

Results section is given clearly and separately for each analysis. However, the limitations of the study were not added. If possible, I think that adding this will add value to the study.

References, a discussion, and a conclusion are sufficient. Additionally, the language is highly fluid and simple to understand. I advise minor revision.
